# Generation of Retinaldehyde for Retinoic Acid Biosynthesis

**DOI:** 10.3390/biom10010005

**Published:** 2019-12-18

**Authors:** Olga V. Belyaeva, Mark K. Adams, Kirill M. Popov, Natalia Y. Kedishvili

**Affiliations:** 1Department of Biochemistry and Molecular Genetics, Schools of Medicine and Dentistry, University of Alabama at Birmingham, Birmingham, AL 35294, USA; kpopov@uab.edu (K.M.P.); nkedishvili@uab.edu (N.Y.K.); 2Stowers Institute for Medical Research, Kansas City, MO 64110, USA; maa@stowers.org

**Keywords:** retinoic acid, retinol, retinaldehyde, short-chain dehydrogenase, vitamin A, retinol dehydrogenase, retinaldehyde reductase, embryogenesis

## Abstract

The concentration of all-*trans*-retinoic acid, the bioactive derivative of vitamin A, is critically important for the optimal performance of numerous physiological processes. Either too little or too much of retinoic acid in developing or adult tissues is equally harmful. All-*trans*-retinoic acid is produced by the irreversible oxidation of all-*trans*-retinaldehyde. Thus, the concentration of retinaldehyde as the immediate precursor of retinoic acid has to be tightly controlled. However, the enzymes that produce all-*trans*-retinaldehyde for retinoic acid biosynthesis and the mechanisms responsible for the control of retinaldehyde levels have not yet been fully defined. The goal of this review is to summarize the current state of knowledge regarding the identities of physiologically relevant retinol dehydrogenases, their enzymatic properties, and tissue distribution, and to discuss potential mechanisms for the regulation of the flux from retinol to retinaldehyde.

## 1. Introduction

All-*trans*-retinoic acid (RA) is produced from all-*trans*-retinol in two oxidative steps: First, retinol is oxidized to retinaldehyde, and then retinaldehyde is oxidized to RA reviewed in ref. [1] (Figure 1). The oxidation of retinol to retinaldehyde is a reversible and a rate-limiting step in the pathway of RA biosynthesis, whereas the oxidation of retinaldehyde is irreversible and occurs at a higher rate than the oxidation of retinol [2]. The enzymes that oxidize retinaldehyde to RA were the first to be defined at the molecular level. Groundbreaking work from the laboratory of Dr. Ursula Dräger in mid-1990s showed that mouse embryos contain three types of retinaldehyde dehydrogenase (RALDH) activity [3,4,5]. The enzymes responsible for these three types of activity were subsequently identified as members of the aldehyde dehydrogenase (ALDH) family of proteins ALDH1A1, 1A2, and 1A3 [6,7,8]. Detection of RALDH activity in embryonic tissues was greatly facilitated by the RA reporter assay developed by Rossant et al. [9]. This assay employed a RA response element (RARE) of the RARβ gene inserted upstream of the promoter from the mouse *hsp*68 gene fused to the gene encoding β-galactosidase [9]. The expression pattern of the resulting RARE-lacZ reporter in mouse embryo overlapped with, but was not identical to, the reported expression pattern of the RA-inducible RARβ gene. Nevertheless, although the specific pattern of RARE-lacZ expression of the *RAREhsplacZ* transgene did not fully coincide with the proposed sites of RA action, this assay allowed for tracking of the changes in RA signaling activity depending on the localized levels of RA at specific sites in the embryo. The assays employing the RARE-lacZ reporter demonstrated that the spatiotemporal pattern of RA signaling undergoes changes on an hourly basis throughout the embryo ([9,10] reviewed in [11]). This dynamic pattern of changes in RA signaling was found to closely correlate with the expression pattern of RALDH2 (ALDH1A2), the primary retinaldehyde dehydrogenase during embryogenesis [10]. However, the source of retinaldehyde for RALDH2-mediated RA biosynthesis remained obscure until 2007, when an *N*-ethyl-n-nitrosourea-induced forward genetic screen linked mutations in the gene encoding retinol dehydrogenase 10 (*Rdh10*) to craniofacial, limb, and organ abnormalities [12]. This phenotype, called RDH10*^trex^*, was caused by the severely reduced ability of mutant RDH10 to oxidize retinol to retinaldehyde, resulting in insufficient RA signaling. RDH10 became the first enzyme among several candidate enzymes displaying a retinol dehydrogenase activity in vitro to be proven to contribute to RA biosynthesis in vivo. Importantly, subsequent analyses of embryos with targeted knockout of the *Rdh10* gene provided evidence for the existence of RDH10-independent sources of retinaldehyde for RA biosynthesis in embryos as well as in adult tissues [12,13,14,15]. This review will focus on our recent findings regarding the enzymes that produce retinaldehyde in vivo; the molecular mechanisms that control the rate of retinaldehyde output to maintain RA homeostasis; and the enzymes that maintain the homeostasis of retinol—the primary source of RA in the cells.

## 2. Retinoid-Active Members of the Short-Chain Dehydrogenase/Reductase Superfamily of Proteins

The short-chain dehydrogenase/reductase (SDR) superfamily of proteins includes enzymes involved in metabolism of lipids, xenobiotics, prostaglandins, steroid hormones, and retinoids [16]. In vivo evidence suggests that members of two families of SDRs are involved in the regulation of RA homeostasis, SDR16C and SDR7C (reviewed in [1,17]). Interestingly, while the enzymes from both families recognize retinol and retinaldehyde as substrates, the two groups display distinct differences in their cofactor preferences: Most members of the SDR16C family (except for DHRS3, see below) exhibit higher binding affinities for NAD(H) as cofactor, whereas members of the SDR7C family prefer NADP(H). If retinol and retinaldehyde are equally available, the cofactor preference largely dictates the direction of the reaction in vivo. In general, the concentration of the oxidized NAD^+^ greatly exceeds that of NADH (1000:1), while the concentration of NADPH exceeds that of NADP^+^ (1:100) [18]. Thus, the NAD(H)-dependent oxidoreductases usually function in the oxidative direction in intact cells, whereas the NADP(H)-dependent enzymes function in the reductive direction.

## 3. Retinol Dehydrogenase 10 (RDH10) and Regulation of the Flux from Retinol to Retinaldehyde

RDH10 belongs to the 16C family of the SDRs (www.sdr-enzymes.org) [16]. The cDNA encoding RDH10 was originally cloned from the human retina cDNA library based on its sequence similarity to retina SDR1 (retSDR1) [19,20,21]. Subsequent studies showed that RDH10 is widely expressed in adult and embryonic tissues.

Enzymatic characterization of recombinant human RDH10 (SDR16C4) carried out by our laboratory demonstrated that RDH10 acts as a high-affinity retinol dehydrogenase with a preference for NAD^+^ as cofactor [22] (Table 1). We also showed that human RDH10 contributes to RA biosynthesis when overexpressed in organotypic skin raft cultures [23], whereas three other candidate enzymes from the 9C family of SDRs, which contributed to RA biosynthesis when overexpressed in cultured cells [24], did not appear to raise the levels of RA in the organ-like setting of epidermal skin rafts [23].

The spatiotemporal pattern of mouse RDH10 expression during embryogenesis was shown to coincide closely with that of RALDH2 [15,25,26]. Mutagenesis and targeted gene knockout studies in mice confirmed that a functional RDH10 is critical for survival until embryonic day 11.5 (E11.5), as *Rdh10^−/−^* embryos could be rescued by maternal supplementation of retinaldehyde from E7.5 to E11.5 [13]. Genetic disruption of murine *Rdh10* gene resulted in a marked reduction in RA synthesis that led to numerous developmental abnormalities. RDH10-deficient embryos displayed defects in axial extension and embryonic turning, abnormal hindbrain and craniofacial patterning, agenesis of posterior pharyngeal arches, perturbed somitogenesis, hypoplastic forelimb buds, and abnormal organogenesis of multiple systems—including heart and vasculature, lungs, and gastrointestinal tract. Embryos carrying a targeted knockout of *Rdh10* died by E12.5 [13,26], while embryos carrying various mutant alleles survived until E13.5–E15.5 [12,27], or until late gestation [28]. Interestingly, testicular cell-specific conditional knockout of *Rdh10* showed that deficiency of RDH10 in both Sertoli and germ cells completely impaired testicular RA signaling in juvenile animals [29]. However, spermatogenesis was progressively recovered in adult *Rdh10* conditional knockout mice, suggesting that RDH10 is not essential for adult spermatogenesis [29].

Concurrently with the studies on RDH10, another member of the SDR16C family, retSDR1 [19], which served as the template for cloning of RDH10, was also found to regulate the levels of RA signaling during embryogenesis [30,31]. This protein, renamed as dehydrogenase reductase 3 (DHRS3, SDR16C1), appeared to act in the direction opposite to RDH10. A knockdown of *dhrs3a* gene expression in zebrafish embryos increased the expression levels of RA-inducible *cyp26a1* gene and RARE-GFP reporter construct in the spinal cord [30]. In mice, targeted knockout of *Dhrs3* resulted in a ~30% increase in RA levels, reduction in the levels of retinol and retinyl esters, and embryonic lethality late in gestation [31,32]. Cumulatively, these observations were consistent with the report that DHRS3 acts as an NADP(H)-dependent retinaldehyde reductase [19]. However, the reported catalytic activity of DHRS3 was much lower than that of other retinaldehyde reductases of the SDR superfamily of proteins [33,34]. This raised a question about the physiological relevance of DHRS3 function as an enzyme.

To resolve this puzzle, we investigated the enzymatic properties of DHRS3. DHRS3 overexpressed in HEK293 cells did not increase the rate of retinol biosynthesis from exogenously added retinaldehyde in living cells. However, silencing of endogenously expressed DHRS3, which was abundantly expressed in HepG2 cells, resulted in an increased rate of retinol conversion to retinaldehyde and RA [32]. This observation suggested that the endogenous DHRS3 acted as a retinaldehyde reductase whereas the ectopically expressed DHRS3 somehow lacked this ability. We hypothesized that since the major source of retinaldehyde in the cells comes from RDH10, the endogenous DHRS3 might have an advantage over the ectopically expressed DHRS3 in terms of access to retinaldehyde. To create an additional source of retinol for the recombinant DHRS3, we co-expressed DHRS3 with recombinant RDH10. Indeed, this experiment allowed us to detect the retinaldehyde reductase activity of DHRS3 for the first time [32] (Figure 2). Further studies revealed that RDH10 and DHRS3 expressed in Sf9 cells form a heterooligomeric membrane-associated complex consisting of at least two subunits of RDH10 and two subunits of DHRS3 [35].

Enzymatic analysis of DHRS3 co-expressed with RDH10 showed that, similarly to other SDRs, DHRS3 acted as a bidirectional oxidoreductase that catalyzed both the reductive and oxidative directions in vitro and preferred NADP(H) as cofactor [32]. The preference for NADP(H) ensured that in living cells DHRS3 would function exclusively in the reductive direction, because NADP^+^ exists mostly in the reduced form. Indeed, when co-expressed with RDH10 in HEK293 cells, DHRS3 converted exogenously supplied retinaldehyde to retinol, and decreased the flux from provided retinol to retinaldehyde and RA, counteracting the oxidative activity of RDH10. The activated DHRS3 exhibited a low *K_m_* value for retinaldehyde (∼0.15 μM) similar to those of other SDRs with retinaldehyde reductase activity, such as retinol dehydrogenase 11 (RDH11) (∼0.12 μM) and retinol dehydrogenase 12 (RDH12) (0.04 μM for purified) [33,34]. The *K_m_* value for NADPH (2.6 µM) measured for DHRS3 in complex with RDH10 was also similar to those of RDH11 (0.47 μM) and RDH12 (0.74 μM). The formation of the complex stabilized both proteins by extending their half-lives. Enzymatic activities of each component in the hetero-oligomeric retinoid oxidoreductase complex (ROC) were analyzed by using enzyme-specific combinations of substrates and cofactors. The activity of the RDH10 component of the ROC was measured using all-*trans*-retinol (or 11-*cis*-retinol) plus NAD^+^. Remarkably, the activity of the complex-bound RDH10 exceeded the activity of RDH10 homo-oligomers several-fold. The activity of DHRS3 when bound to RDH10 was measured using all-*trans*-retinaldehyde plus NADPH; the activity of DHRS3 homo-oligomers was essentially undetectable [32].

The expression of DHRS3 is known to be induced by RA [36,37]. To determine the physiological implications of the rise in DHRS3 protein content on the flux from retinol to retinaldehyde, we mimicked the conditions of increasing DHRS3 protein by co-transfecting HEK293 cells with a fixed amount of RDH10-FLAG construct and increasing amounts of DHRS3-FLAG construct [35]. The control cells were transfected only with RDH10-FLAG. To our surprise, even at the lowest dose, the presence of DHRS3 completely abolished the increase in RA biosynthesis caused by overexpression of RDH10, and further increase in DHRS3 protein had little or no effect on the rate of RA production from retinol. This observation suggested that the complex formed by RDH10 and DHRS3 had a homeostatic effect on RA biosynthesis. Indeed, a reverse experiment where RDH10 protein levels were gradually increased while DHRS3 protein was kept relatively constant (increasing slightly due to the stabilizing effect of RDH10) showed that the rise in RA levels following the growing amount of RDH10 protein was considerably blunted compared to the cells that expressed RDH10 without DHRS3. These studies demonstrated the importance of the closed circuit of retinol/retinaldehyde interconversion within the ROC composed of enzymes with antagonistic activities, RDH10 and DHRS3, for the maintenance of RA homeostasis (Figure 2). In essence, the formation of the ROC made the rate of RA biosynthesis largely independent of the concentration of ROC components. Disruption of the ROC-generated circuit by a knockdown of DHRS3 in HepG2 cells resulted in an increased flux through the RA biosynthesis pathway and elevated RA levels despite the overall decrease in RDH10 protein, which was destabilized in the absence of DHRS3. Thus, the bifunctional nature of ROC provided the RA-based signaling system with robustness by safeguarding appropriate RA concentration despite naturally occurring fluctuations in RDH10 and DHRS3.

The bifunctional complex of RDH10 and DHRS3 presents an example of a “paradoxical component” (reviewed in [38]). The key feature of the paradoxical component, which ensures the robustness of the metabolic output, is that monofunctional enzymes must function as a complex to carry out opposing reactions. RDH10 and DHRS3 within the antagonistically bifunctional ROC simultaneously catalyze oxidation of retinol and reduction of retinaldehyde, thus exerting opposing effects on the levels of RA precursor retinaldehyde and, subsequently, RA. The mutually activating interaction between the two enzymes means that an increase in the concentration of either component increases both the oxidation of retinol and the reduction of retinaldehyde, thus canceling out the effect on the steady-state output of RA. In contrast, a retinol dehydrogenase and a retinaldehyde reductase functioning as independent enzymes would make the output of retinaldehyde and RA dependent on the concentrations of both proteins in the circuit.

Studies in animals demonstrated that the retinol dehydrogenase activity of membrane fractions isolated from *Dhrs3^−^*^/*−*^ mouse embryos was decreased two-fold [32], but despite the reduced RDH activity, the RA levels in these embryos were ~30% higher than in WT embryos [31,32,39]. The reduced in vitro RDH activity in *Dhrs3^−/−^* membranes may have resulted from the loss of stabilizing and activating effect of DHRS3 on its RDH10 partner in ROC. The decreased levels of RDH10 protein and increased levels of RA that we observed in *DHRS3-*silenced human HepG2 cells [32] also support this notion. The fact that, despite its reduced levels in the absence of DHRS3, RDH10 was able to maintain higher steady-state levels of RA illustrates the significance of a circuit created by the antagonistic activities of the ROC. These observations together with the fact that DHRS3 is inactive as a retinaldehyde reductase in the absence of RDH10, led us to conclude that the primary role of DHRS3 was to control the retinol dehydrogenase activity of RDH10 rather than to serve as a general purpose retinaldehyde reductase.

Models of reduced DHRS3 functionality, such as *Dhrs3^−/−^* mouse embryos [31,32,39], human HepG2 cells with shRNA-mediated knockdown of *DHRS3* [32], or expression of catalytically inactive DHRS3 Y188A mutant in cell lines, which cannot reduce retinaldehyde to retinol, but still binds to RDH10 and activates it, augmenting RA biosynthesis [32,35], illustrate the metabolic consequences of a malfunctioning homeostatic circuit, where the tight control over RA output normally imposed by the bifunctional complex is lost. The ablation of DHRS3 leads to an increased flux from retinol to RA, numerous changes in cultured cells and lethality in knockout embryos. The defects in *Dhrs3^−/−^* mouse embryos were consistent with elevated RA levels, including abnormalities in axial skeletal (vertebral fusion) and craniofacial development (cleft palate), and anomalies in heart and coronary vasculature formation ([31,32,40], reviewed in [41]). DHRS3 loss-of-function studies both in zebrafish and *Xenopus* [42] also resulted in phenotypes consistent with overproduction of RA, including aberrant hindbrain patterning in zebrafish *dhrs3a* morphants [30] and defects in the patterning of neuroectoderm, axial extension, and somitogenesis in frog, suggesting a conservation of DHRS3 function.

Studies in zebrafish have underscored the importance of a robust RA morphogen gradient during embryogenesis and identified a negative feedback loop involving the RA-degrading enzyme cyp26a1 and cellular RA-binding protein type 2 as a mechanism for increased robustness [43,44,45]. Our findings show that the antagonistically bifunctional ROC provides an additional mechanism to ensure the robustness of RA output during embryogenesis, which functions at the level of RA precursor biosynthesis, and utilizes a principle different from the negative feedback loop at the level of RA degradation.

DHRS3 and RDH10 are closely related [1], sharing 54% sequence identity. Both proteins are also highly conserved, with the mouse orthologs sharing ~98% sequence identity with their human counterparts. Thus, the activating interaction between the two proteins may be a regulatory mechanism conserved across species.

## 4. RDH10-Related Dehydrogenases

Although RDH10 was undoubtedly the major retinol dehydrogenase during embryogenesis, RDH10-null embryos could be rescued with maternal retinaldehyde supplementation during E7.5–E11.5 and survived through adulthood after the supplementation was discontinued [13], thus suggesting the existence of additional sources of retinaldehyde at later embryonic stages. Furthermore, measurements of the total embryonic RDH activity showed that *Rdh10^trex^* embryos retained some NAD^+^-dependent membrane-bound RDH activity, albeit greatly reduced in comparison to WT embryos [14]. Interestingly, it was observed that the NADP^+^-dependent activity of *Rdh10^trex^* embryonic membranes was also decreased at E12 and E13 [14]. This could be in part due to the loss of the NADP^+^-dependent DHRS3 activity in the absence of the activating and stabilizing effects of RDH10 in *Rdh10^trex^* mutant mice. Studies using LacZ reporter have pinpointed the areas of residual RA synthesis in RDH10-deficient embryos—in particular, in the neural tube and lateral mesoderm (Figure 3) [13,26,46]—providing further evidence for the existence of yet unidentified retinol dehydrogenases.

Which enzyme(s) could be responsible for the residual RA biosynthesis in RDH10-null embryos at E10 and later? Our previous studies suggested that in *Xenopus laevis*, retinol dehydrogenase epidermal 2 (rdhe2), structurally and evolutionary related to RDH10, contributes to RA biosynthesis during embryonic development [47]. Frog rdhe2 exhibits an excellent activity toward all-*trans*-retinol, comparable with that of human RDH10 when expressed in the same cell culture system. Embryonic expression pattern of frog *rdhe2* was found to overlap with some of the reported domains of rdh10 expression [48]; namely, forebrain, midbrain, otic vesicle, and eye. However, *rdhe2* expression domains in some of these areas appeared to be broader than those of *rdh10*. Unlike *rdh10*, *rdhe2* was present in notochord. *rdhe2* was also highly expressed in the brain at late tailbud and tadpole stages but was absent from the developing head earlier. This dynamic expression pattern of *rdhe2* was consistent with the reports that RA synthesis was largely excluded from the head at early stages to allow for proper development of anterior structures and appeared later at several sites of the developing brain [11,49,50]. In *rdhe2* morphants, changes in the expression of RA-responsive posterior hindbrain marker *hoxd4* indicated that rdhe2 affected RA signaling required for establishing segmental identity of the hindbrain. Thus, in some areas of developing frog embryo, such as notochord and hindbrain, rdhe2 might have been largely responsible for the generation of retinaldehyde.

Based on the studies in *Xenopus*, we concluded that mammalian orthologs of frog rdhe2 might also function as retinol dehydrogenases and pursued further analysis of their in vivo function in mice. Mice have two genes orthologous to rdhe2, one encoding retinol dehydrogenase epidermal 2 (RDHE2, SDR16C5), and another one encoding RDHE2-similar (RDHE2S, SDR16C6). Our recent analysis of the substrate and cofactor specificity of murine RDHE2 and RDHE2S showed that RDHE2S exhibits a higher catalytic activity than RDHE2, and that both enzymes act in the oxidative direction when overexpressed in cultured cells, in agreement with their preference for NAD^+^ as cofactor [51] (Table 1). Interestingly, although RDHE2 and RDHE2S belong to the same SDR16C subfamily of SDRs as RDH10 [16] and are the most closely related to RDH10 proteins, they do not appear to interact with DHRS3. Co-expression of human RDHE2 or murine RDHE2S with human DHRS3, which is 97.7% identical to murine DHRS3, in HEK293 cells had no activating effect on DHRS3 [52]. Pull-down assays using RDH10 protein as a bait failed to co-immunoprecipitate human RDHE2 or human retinol dehydrogenase 11 (RDH11), an enzyme from a different family of SDRs (SDR7C1) [16,33], further indicating that the protein–protein interaction between RDH10 and DHRS3 is highly specific for this pair of proteins [35].

To directly address the in vivo function of murine RDHE2 and RDHE2S, we generated double knockout (DKO) mouse strains [51]. Although both RDHE2 and RDHE2S were found to be expressed in the embryo, DKO pups were born at Mendelian frequency, did not show obvious developmental abnormalities, survived to adulthood, and were fertile. However, DKO mice exhibited histological changes in skin, altered hair follicle cycle, and enlarged sebaceous and meibomian glands, which was consistent with a decrease in RA levels. In adult mice, skin was found to be the major expression site of both RDHE2 and RDHE2S [51]. Analysis of membrane fractions from DKO skin showed that together RDHE2 and RDHE2S account for up to 80% of the total membrane-associated retinol dehydrogenase activity. This demonstrated that, while present in skin, RDH10 was not the major retinol dehydrogenase in this tissue. At the molecular level, analysis of gene expression revealed an upregulation of markers of sebocyte differentiation and hair follicle stem cells, and effectors of Wnt/β-catenin pathway in DKO mice, also consistent with the downregulation of RA signaling. The described phenotypic features were present in mice maintained on chow diet as well as mice fed a vitamin A-deficient (VAD) diet.

Interestingly, while RDHE2S exhibited a higher retinol dehydrogenase activity than RDHE2, a single *Rdhe2s* gene knockout did not display the same phenotype as the DKO mice [51]. *Rdhe2s^−/−^* mice had normal eyelids and regular hair growth. This observation indicated that first, despite its lower enzymatic activity, RDHE2 was essential for RA biosynthesis in mice; and second, the functions of these two enzymes may be redundant because of the overlapping expression pattern. The reason for such an overlap is unclear at this time.

In humans, *RDHE2* gene encodes a functional enzyme with a retinol dehydrogenase activity [53], whereas *RDHE2S (SDR16C6)* appears to have lost its function and is classified as pseudogene [54]. It is possible that human RDHE2, which exhibited a wider tissue distribution pattern than either one of murine orthologs [51] fulfills the mission of both enzymes. Like murine RDHE2, human RDHE2 (SDR16C5) also displayed a rather low activity when assayed as a partially purified enzyme in vitro [54], possibly suggesting the existence of a partner protein that regulates its activity in vivo. Interestingly, human RDHE2 appears to be regulated at the transcriptional level by RA via a negative feedback loop [54].

Thus, RDHE2 and RDHE2S are physiologically relevant retinol dehydrogenases in mammals, which despite their importance for tissue-specific RA biosynthesis are not critical for survival during mouse embryogenesis. It is worth noting that VAD diet administration in our study was used to model adult vitamin A deficiency. This regiment was selected in order to avoid embryonic vitamin A deficiency and allow for normal development until weaning. Therefore, it remains to be investigated whether DKO embryos may be more sensitive to inadequate vitamin A supply during development, or whether their contribution to embryonic RA synthesis becomes more critical on the background of partially inactivating mutations of RDH10. The genome-wide association studies have linked the chromosomal region harboring *RDHE2* (SDR16C5) to stature and growth in cattle, humans, and pigs [55,56,57,58,59,60,61,62], and beak deformity in chickens [63]. *RDHE2* was also identified as the important candidate gene in pig growth trait by an integrative genomic approach [64]. These observations indicate that while apparently redundant during normal embryogenesis, the roles of RDHE2 and RDHE2S in development merit further investigation.

## 5. Retinol Dehydrogenase 11 and Maintenance of Retinol Homeostasis

DHRS3 controls the RA output during development by counteracting the synthesis of its precursor, all-*trans*-retinaldehyde. Thus, the question arises whether other enzymes with retinaldehyde reductive activities can also affects embryonic RA synthesis. We will focus on retinoid-active membrane-bound enzymes of SDR7C family [16], which prefer NADP(H) cofactors, and therefore are likely to function as reductases in vivo.

The founding member of the SDR7C family was RDH11 (also known as SDR7C1, RalR1, PSDR1). This enzyme was first identified at the level of transcription in human prostate [65]. Biochemical analysis of the encoded protein showed that it exhibits high catalytic activity towards all-*trans*-retinaldehyde (Table 2) [66]. The mouse ortholog of RDH11 was independently cloned from liver as a transcript regulated by sterol response element binding proteins [67]. In vitro, both human and mouse enzymes were highly active toward retinaldehyde, but mouse RDH11 also showed significant activity toward the reduction of medium chain lipid-derived aldehydes [67]. Activity toward retinoids, expression in the eye and similarity to photoreceptor-specific RDH12 [1] led to extensive studies of RDH11′s role in the visual cycle and delayed dark adaptation. However, these studies did not reach a consensus on RDH11′s contribution to the regeneration of visual chromophores [68,69] and ruled out its role in protection against toxic lipid peroxidation products in the retina [70].

During development, *Rdh11* transcript was detected at a constant level in the eye at E12 and later stages [71]. More recent RNA sequencing-based analysis suggested a broader expression in embryonic tissues including stage E8.5 [72,73]. However, unlike *Dhrs3* knockout, *Rdh11^−/−^* embryos were viable and developed normally. With the lack of embryonic phenotype, the role of RDH11 in metabolism of retinoids remained unclear until our recent study produced evidence that RDH11 functions as an all-*trans*-retinaldehyde reductase in vivo and is essential for the maintenance of all-*trans*-retinol steady-state levels in mouse liver and testis [74]. This study showed that the levels of retinol in livers and testes of *Rdh11*^−/−^ mice maintained on VAD diet were reduced by approximately one-third in comparison to wild type littermates. The levels of retinol in testes were also lower in *Rdh11^−/−^* males compared to WT littermates when mice were bred on retinol binding protein 4 (RBP4)-deficient background (*Rdh11*^−/^^−^; *Rbp4*^−/−^) and received β-carotene as the sole source of vitamin A. In the absence of serum RBP4, retinol was not redistributed from liver, and peripheral tissues were reliant on dietary β-carotene as the source of vitamin A. Our results demonstrated that testis could utilize circulating β-carotene to supplement its local retinoid stores and that RDH11 contributed to β-carotene processing in peripheral tissues. Thus, RDH11 appeared to be critical for the maintenance of physiological levels of retinol under the conditions of restricted vitamin A supply (Figure 4).

To test whether *Rdh11* knockout may exacerbate the effect *Dhrs3* ablation on RA synthesis in the embryo, we crossed *Rdh11*^−/−^ and *Dhrs3*^−/−^ strains to obtain double knockout embryos, as well as single knockouts on the same mixed genetic background [39]. The RA levels in *Rdh11*^−/−^; *Dhrs3*^−/−^ embryos were higher than in WT or *Rdh11*^−/−^ embryos, but there was no further increase in RA levels or earlier lethality observed in double knockout embryos in comparison to *Dhrs3*^−/−^ embryos. This result showed that—unlike DHRS3—RDH11 does not play a significant role in the maintenance of RA homeostasis in fetal tissues under conditions of sufficient vitamin A supply. There was also no further decrease detected in the levels of free retinol or retinyl ester stores in vitamin A-sufficient *Rdh11*^−/−^; *Dhrs3*^−/−^
*versus Dhrs3*^−/−^ embryos. Unexpectedly, retinyl ester stores were elevated in *Rdh11*^−/−^ embryos in comparison to WT. There were no compensatory changes detected in the expression of other retinaldehyde reductases or genes promoting retinol uptake and esterification in *Rdh11*^−/−^ group, which suggested that the effect may be maternal, as most of the embryos in this group were obtained from *Rdh11*^−/−^ mothers. The maternal nature of the effect was confirmed in a follow-up experiment, where all *Rdh11*^−/−^ and WT embryos were obtained as littermates from *Rdh11* heterozygote dams, and no difference in retinoid stores was observed between these groups. This indicated that while *Rdh11* status of the embryo had no effect on fetal retinoid levels under normal vitamin A supply, *Rdh11* status of the female breeder may have affected the transfer of retinyl esters across the placenta, or the availability of transferred free retinol for in situ esterification in the embryo.

The results of our studies demonstrated that although DHRS3 and RDH11 enzymes catalyze the same chemical reaction in vivo, the reduction of retinaldehyde to retinol, their physiological functions are different and not overlapping. While DHRS3 is involved primarily in the control of RA homeostasis as a reductive partner in the bifunctional oxidoreductase complex, RDH11 appears to be required for the homeostasis of retinol in specific tissues under conditions of restricted vitamin A supply.

## 6. RDH11-Related Enzymes

Retinol dehydrogenases 12–14 (RDH12–14) are NADP(H)-preferring enzymes phylogenetically related to RDH11. RDH12 is a photoreceptor-specific protein and thus is expected to function exclusively in the eye. RDH13 and RDH14 are expressed ubiquitously in human and mouse tissues. Mouse transcriptome analysis by ENCODE [73] revealed an overlapping expression of *Rdh13* and *Rdh14* in adult and embryonic tissues, including the central nervous system at E11.5. Analysis of early craniofacial gene expression detected transcripts of *Rdh13* and *Rdh14* even earlier (E8.5) [73]. Cap analysis of gene expression data generated by the FANTOM5 project suggested a broad expression pattern of both enzymes in embryonic tissues beginning at E11 [78].

Our previous studies have characterized both human RDH13 and RDH14 as highly efficient all-*trans*-retinaldehyde reductases (Table 2) [79,80]. RDH13 differs from the other members of SDR7C family in that it localizes to the inner membrane of mitochondria rather than endoplasmic reticulum and exhibits a higher *K_m_* value for retinaldehyde reduction [80,81]. The localization of RDH13 in the inner membrane of mitochondria facing an intermembrane space prompted us to speculate that it may function as a detoxifying enzyme, which serves as a barrier protecting the mitochondria against the highly reactive retinaldehyde. It has been also proposed that RDH13 may work in concert with mitochondrial β-carotene-9′,10′-dioxygenase type 2 (BCDO2), which acts as a scavenger of carotenoids by cleaving them into apocarotenoids [82]. Hence, the function of RDH13 could be to reduce apocarotenals generated by BCDO2 to corresponding alcohols, thus removing reactive aldehyde groups of the primary cleavage products.

Mouse knockout of *Rdh13* gene generated by Wang et al. [83] did not result in embryonic lethality, indicating that RDH13 is not essential for embryogenesis under normal conditions. Examination of the retina architecture and visual function revealed that retinas of *Rdh13^−/−^* mice were more susceptible to light-induced damage. Prolonged exposure to intense light led to thinning of the retina and mitochondrial damage, manifested as swollen organelles with disrupted cryptae. Upregulation of the markers of mitochondrial apoptosis pathway observed in the light-exposed *Rdh13^−/−^* retinas was consistent with the suggested protective role of RDH13 against reactive aldehydes. At the same time, RDH13-deficient mice were less susceptible to carbon tetrachloride-induced liver fibrosis [84,85]. RDH13 deficiency alleviated liver injury, inhibited activation of hepatic stellate cells and reduced apoptosis. The effect of RDH13 knockout on retinoid metabolism or RA signaling was not assessed in any of these studies, and therefore it remains unknown whether the retinaldehyde reductase activity of RDH13 was responsible for the observed differences (Figure 4).

Human and mouse RDH14 proteins are 90% identical. This degree of conservation between species is not as high as that of RDH10 and DHRS3 (99% and 95% identity, respectively, between human and mouse enzymes), but still suggests an important common function. Like human RDH14, the mouse enzyme was shown to effectively reduce all-*trans*-retinaldehyde and was also active toward *cis*-retinaldehydes [86]. Mouse RDH14 enzyme was also shown to catalyze the aldehyde-alcohol redox coupling reaction, a proposed unconventional mechanism of chromophore regeneration in cones [87]. Nevertheless, an animal knockout of *Rdh14* has not yet been characterized, and the in vivo role of RDH14 has not yet been established (Figure 4).

## 7. Future Studies

Recent progress in enzymology of retinoid metabolism has greatly advanced our understanding of vitamin A action while simultaneously highlighting gaps in our knowledge and identifying directions for future research. Identification of RDH10 and DHRS3 as enzymes critical for the regulation of embryonic RA levels in vivo has proved that the oxidation of retinol to retinaldehyde is an important level in the control of RA homeostasis. The discovery that RDH10 and DHRS3 form a bifunctional ROC as a molecular mechanism for such control raises a question whether ROC functions to establish a robust spatiotemporal pattern of RA production in the embryo.

While RDH10 is clearly responsible for most of the retinol oxidation during development, the source of retinaldehyde for the residual RA biosynthesis in RDH10-deficient embryos and its significance presents another open question. Previous studies have ruled out alcohol dehydrogenases [91] and cytochrome P450 CYP1B1 [46] as the enzymes responsible for RDH10-independent RA production in the embryo. Considering that RDHE2 and RDHE2S enzymes contribute to RA generation in adult animals [51], they should be evaluated as potential candidates that may also supply retinaldehyde in a tissue- or stage-specific pattern in the embryo.

As has been pointed out above, *Dhrs3^−/−^* embryos retain significant membrane-associated retinaldehyde reductive activity. Studies in *Rdh11* knockout mice [74] have established that RDH11 is required for the homeostasis of retinol in adult testis and the liver, but RDH11 does not appear to affect the retinol levels in the embryo. Figure 5 summarizes the current state of knowledge about the enzymes that were shown to metabolize retinol to retinaldehyde and convert retinaldehyde to retinol in mouse gene knockout models. The other components of the retinoid network have been thoroughly described in previous comprehensive reviews by the experts in the field [41,75,76,77,92]. Future studies are necessary to evaluate the contribution of RDH11-related enzymes RDH13 and RDH14 to retinoid metabolism in adult and embryonic tissues, and to uncover additional retinaldehyde reductases and retinol dehydrogenases that are critical for fine-tuning retinol and RA levels in a tissue-specific manner and under changing physiological conditions, such as restricted vitamin A supply.

## Figures and Tables

**Figure 1 biomolecules-10-00005-f001:**
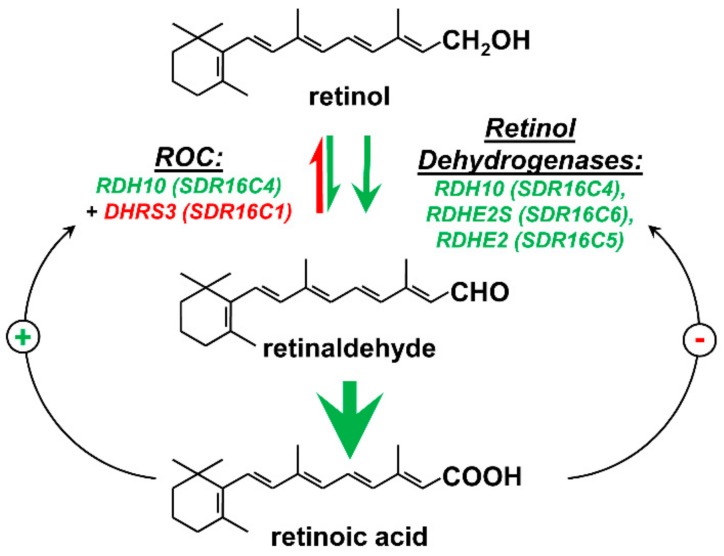
Enzymes that control the generation of retinaldehyde in vivo. Green arrows indicate the oxidative nicotinamide adenine dinucleotide (NAD^+^)-dependent reactions. The red arrow indicates the reductive nicotinamide adenine dinucleotide phosphate (NADPH)-dependent direction. Black arrows represent the retinoic acid—mediated transcriptional feedback loops that upregulate (+) the expression of the reductive enzyme DHRS3 and downregulate (−) the expression of the oxidative enzyme RDHE2 in response to an increase in retinoic acid levels. ROC stands for the retinoid oxidoreductive complex composed of RDH10 and DHRS3.

**Figure 2 biomolecules-10-00005-f002:**
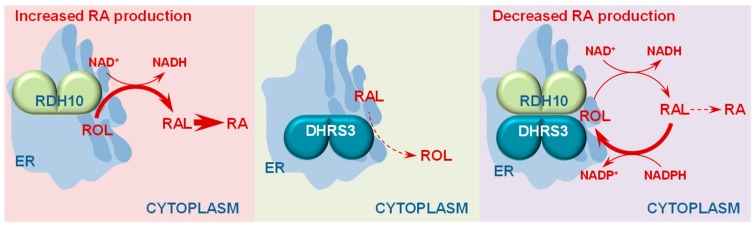
The concept of antagonistically bifunctional retinoid oxidoreductase complex. RDH10 expressed in cells functions as an NAD^+^-dependent enzyme that catalyzes the oxidation of retinol (ROL) to retinaldehyde (RAL). DHRS3 has little or no enzymatic activity on its own; however, when co-expressed with RDH10, DHRS3 displays a robust retinaldehyde reductase activity in the presence of NADPH as cofactor. Protein affinity enrichment studies indicate that RDH10 and DHRS3 physically bind to each other forming a heterooligomeric retinoid oxidoreductive complex (ROC) that interconverts retinol and retinaldehyde utilizing NAD^+^ or NADPH as cofactors. The ROC contains at least two subunits of each, RDH10 and DHRS3, suggesting that both proteins form homodimers [34]. In the ROC, RDH10, and DHRS3 activate and stabilize each other. This results in a homeostatic effect on the output of retinaldehyde, and hence retinoic acid (RA), which becomes largely independent of the overall RDH10 and DHRS3 protein levels. The induction of DHRS3 gene transcription by retinoic acid may serve to drive the binding equilibrium between RDH10 and ROC towards increased formation of ROC and resetting of the baseline retinoic acid levels.

**Figure 3 biomolecules-10-00005-f003:**
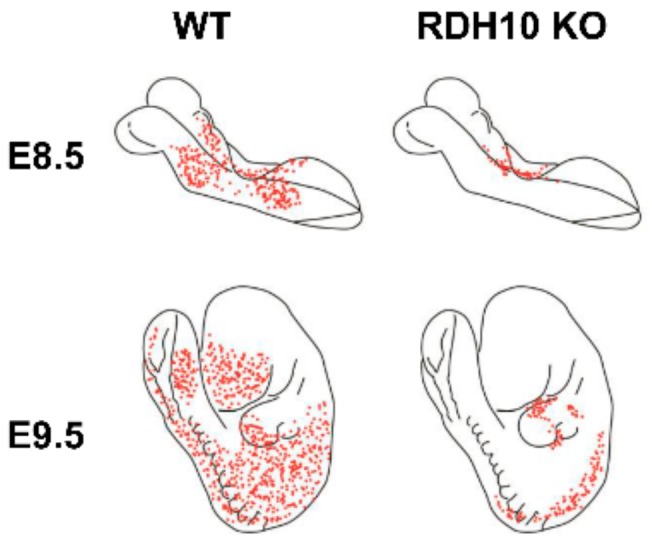
Diagram of sites of residual retinoic acid biosynthesis in RDH10-null embryos. The diagram is based on [13,26,46].

**Figure 4 biomolecules-10-00005-f004:**
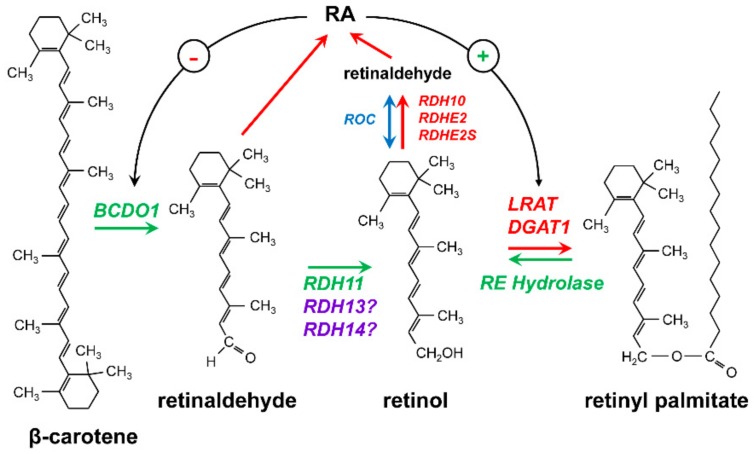
Enzymes that control the homeostasis of retinol. Retinol is generated in the cells either from β-carotene or from retinyl esters. Reactions leading to generation of retinol are represented by green arrows. Reactions that reduce the levels of retinol are indicated by red arrows. β-Carotene undergoes a symmetrical cleavage by β-carotene dioxygenase type 1 (BCDO1) to produce two molecules of all-*trans*-retinaldehyde (reviewed in [75]), which are then reduced to all-*trans*-retinol by RDH11 in liver and testis [74]. The roles of RDH13 and RDH14 in the generation of retinol remain to be established. Retinaldehyde produced from β-carotene can be directly oxidized to retinoic acid (RA). RA regulates the expression levels of BCDO1 via a negative (−) feedback loop [75]. Retinol is produced from retinyl esters by the action of hydrolases (reviewed in [76]). The conversion of retinol to retinyl esters is catalyzed by lecithin-retinol acyltransferase (LRAT) and diacylglycerol acyltransferase 1 (DGAT1) (reviewed in [77]). LRAT expression is upregulated by RA (+). The oxidation of retinol to retinaldehyde is controlled by the NAD^+^-dependent retinol dehydrogenases (RDH10, RDHE2, RDHE2S) and the retinoid oxidoreductase complex (ROC) composed of the NAD^+^-dependent RDH10 and the NADPH-dependent DHRS3.

**Figure 5 biomolecules-10-00005-f005:**
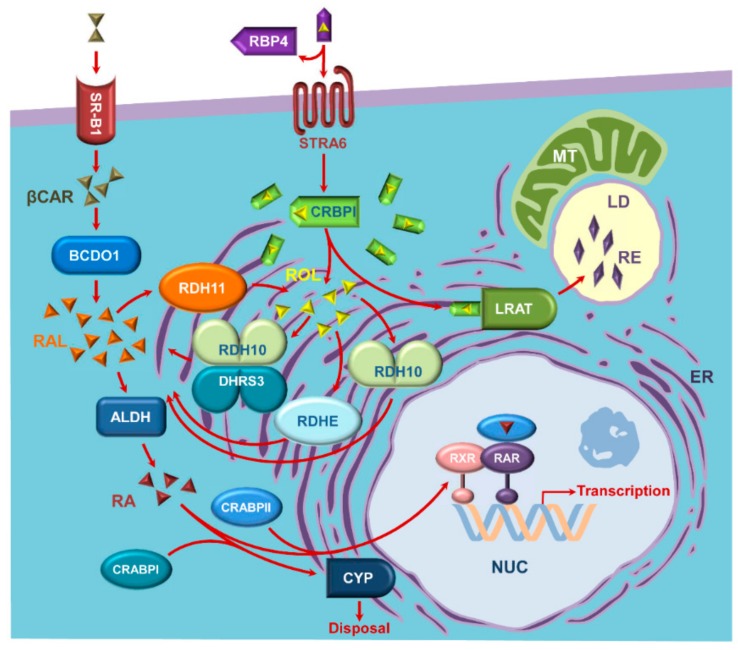
An updated view of the components of the retinoid metabolic and signaling network. Retinol (ROL, depicted as yellow pyramids) is delivered to extrahepatic cells bound to plasma retinol binding protein 4 (RBP4). HoloRBP4 binds to RBP4 receptor STRA6 in STRA6-expressing cells. Most of the cells take up retinol in the absence of STRA6 [88]. Cellular retinol binding protein type I (CRBPI) accepts retinol from STRA6 in the cytoplasm and delivers retinol to membranes of endoplasmic reticulum (ER), where retinol is either esterified by lecithin retinol acyl transferase (LRAT) to retinyl esters (RE, depicted as purple rhombus) or oxidized by retinol dehydrogenases 10 (RDH10) and epidermal retinol dehydrogenases 2 (RDHE2) and RDHE2-similar (RDHE2S) to retinaldehyde (RAL, depicted as orange pyramids). The baseline levels of retinaldehyde are controlled by the retinoid oxidoreductive complex (ROC) composed of RDH10 and DHRS3. Both RDH10 and DHRS3 are found in subcellular microsomal, mitochondrial, and lipid droplet fractions [35,89,90]. Retinaldehyde is oxidized further to retinoic acid (RA, depicted as brown pyramids) by cytosolic aldehyde dehydrogenases of the ALDH1A family (ALDH). Retinoic acid binds to cellular retinoic acid binding proteins (CRABPs) type I or type II and is transferred by holoCRABPII to nucleus (NUC) for binding to heterodimers of retinoic acid receptors (RAR and RXR) or delivered to cytochrome P450 enzymes (CYP) by holoCRABPI for degradation. In addition, β-carotene (βCAR, depicted as a duplicate of olive pyramids) is taken up by the cells through scavenger receptor class B, type 1 (SR-B1), and cleaved into two molecules of retinaldehyde by β-carotene dioxygenase type 1 (BCDO1). Retinaldehyde derived from β-carotene may be oxidized to retinoic acid as described above or converted to retinol by retinol dehydrogenase 11 (RDH11). Other abbreviations: LD, lipid droplet; MT, mitochondria.

**Table 1 biomolecules-10-00005-t001:** Kinetic parameters of SDR16C retinol dehydrogenases. Kinetic constants for each enzyme were determined using the same preparation of microsomes containing the corresponding enzyme. Hence, the rates with different substrates and cofactors for the same enzyme can be compared. UD, undetectable under the conditions of the assay; N.D., not determined. The *K*_m_ values for human RDHE2 were not determined due to the low activity of the enzyme. References are indicated by superscripts.

SDR	Substrate/Cofactor	Apparent *K*_m_ μM	Apparent *V*_max_ nmol⋅min^−1^⋅mg^−1^	*V*_max_/*K*_m_
*Xenopus* rdhe2^51^	all-*trans*-retinol	0.6 ± 0.1	19.5 ± 0.6	32.5
	NAD^+^	108 ± 27	21 ± 1	
	all-*trans*-retinaldehyde	0.6 ± 0.1	3.6 ± 0.1	6
	NADH	8.4 ± 1.7	4.1 ± 0.2	
	11-*cis*-retinol	3.3 ± 0.4	3.9 ± 0.2	1.8
Murine RDHE2^51^	all-*trans*-retinol	UD	UD	
	all-*trans*-retinaldehyde	N.D.	N.D.	
Murine RDHE2S^51^	all-*trans*-retinol	0.87 ± 0.21	8.7 ± 0.6	10
	NAD^+^	460 ± 30	5.8 ± 0.2	
	all-*trans*-retinaldehyde	0.6 ± 0.1	3.6 ± 0.1	6
	NADH	11 ± 3	2.6 ± 0.1	
	11-*cis*-retinol	0.86 ± 0.14	1.34 ± 0.07	1.6
	9-*cis*-retinol	UD	UD	
Human RDHE2^53^	all-*trans*-retinol	N.D.	~0.06	
	NAD^+^	Preferred		
Human RDH10^22^	all-*trans*-retinol	0.035 ± 0.010	1.30 ± 0.05	37
	NAD^+^	100 ± 10	1.30 ± 0.04	
	11-*cis*-retinol	0.06 ± 0.01	1.42 ± 0.06	24
	9-*cis*-retinol	0.04 ± 0.01	0.77 ± 0.05	19

**Table 2 biomolecules-10-00005-t002:** Kinetic constants of human SDR7C enzymes. Kinetic analysis of enzymes expressed in Sf9 microsomes was performed using the same preparations of microsomes containing the corresponding enzyme. Hence, the rates with different substrates and cofactors for the same enzyme are directly comparable. For purified enzymes, *k*_cat_ values were calculated based on the molecular mass of the corresponding enzyme monomer. *V*_max_ values are listed for comparison of purified enzymes’ activities with Sf9 microsomal preparations of the enzymes. References are indicated by superscripts. Abbreviations are: a*t*ROL, all-*trans*-retinol; a*t*RAL, all-*trans*-retinaldehyde; MS, microsomes.

SDR	Substrate/Cofactor	Apparent *K*_m_ μM	Apparent *V*_max_ nmol·min^−1^·mg^−1^	*V*_max_/*K*_m_	*k*_cat_ min^−1^	*k*_cat_/*K*_m_ min^−1^·μM^−1^
RDH11Sf9 MS^33^	all-*trans*-retinol	0.70 ± 0.04	13.2 ± 0.3	19		
NADP^+^	0.4 ± 0.2	8.7 ± 0.8			
all-*trans*-retinaldehyde	0.20 ± 0.01	41 ± 1	205		
NADPH	0.48 ± 0.02	44.0 ± 0.6			
RDH11Purified^33^	all-*trans*-retinol	0.6 ± 0.1	300 ± 20	500	11	18
NADP^+^	1.0 ± 0.1	250 ± 4			
all-*trans*-retinaldehyde	0.12 ± 0.1	506 ± 13	4217	18	150
NADPH	0.47 ± 0.04	530 ± 10			
RDH12,Sf9 MS^34^	NADP^+^ (with a*t*ROL)	1.2 ± 0.2	79.0 ± 2.5			
NADPH (with a*t*RAL)	1.2 ± 0.3	98.0 ± 2.0			
RDH12Purified^34^	all-*trans*-retinol	0.40 ± 0.10	748	1870	27 ± 2	68
11-*cis*-retinol	0.16 ± 0.03			7 ± 0.7	44
9-*cis*-retinol	0.16 ± 0.03			7 ± 0.3	44
NADP^+^ (with a*t*ROL)	3.2 ± 0.2	770 ± 30			
all-*trans*-retinaldehyde	0.04 ± 0.01	997	24,930	36 ± 1.7	900
11-*cis*-retinaldehyde	0.1 ± 0.05			45 ± 4	450
9-*cis*-retinaldehyde	0.14 ± 0.03			14 ± 1	100
NADPH (with a*t*RAL)	0.74 ± 0.06	1020 ± 20			
RDH13Purified^80^	all-*trans*-retinol	~ 3	~ 5			
NADP^+^	~6000	~25			
all-*trans*-retinaldehyde	3.2 ± 0.7	230 ± 24	72	8.2	2.6
NADPH	1.5 ± 0.1	230 ± 24			
RDH14Sf9 MS^79^	all-*trans*-retinol	0.4 ± 0.07	31 ± 2	78		
NADP^+^	0.65 ± 0.11				
all-*trans*-retinaldehyde	0.08 ± 0.2	27 ± 1	338		
NADPH	0.32 ± 0.04

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
