# Peer review of "Generation of Retinaldehyde for Retinoic Acid Biosynthesis"

_biomolecules, 2019, doi:10.3390/biom10010005_

Round 1

Reviewer 1 Report

Manuscript #biomolecules-657233 reviews the state of the art on the generation of retinaldehyde for retinoic acid synthesis. This well written and highly informative manuscript deserves publication with no major changes. Nevertheless, the reviewer would like to raise 3 minor points:

1) page 3, line 102: The outcome of Rdh10 ablation in testicular cells should be mentioned (Tong et al., 2013 Proc. Natl. Acad. Sci USA 110(2):543-8).

2) page 176, line 175: ROC is not defined in the main text.

3) page 8, line 317: Legend of Figure 4 indicate that “HoloRBP4 binds to RBP4 receptor STRA6”. This is true in STRA6-expressing cells only, which are rather the exception in the whole body (e.g. retinal pigmented epithelium cells). A gene knockout study published in 2013 indicates that most of the cells actually take up HoloRBP4 in the absence of STRA6 (Berry et al., 2013 J. Biol. Chem. 288(34):24528-39).

Reviewer 2 Report

This is a very valuable review on a topic that has been infrequently covered. The writing style is clear and overall the review is a pleasure to read e.g. the paradoxical component of RDH10 and DHRS3 acting in a complex is an intriguing discovery.  I just have minor comments to be considered:

The importance of balance of cofactors (NAD, NADH etc) is very interesting and it would be useful to know whether these vary between tissues to influence rates. It would be helpful to discuss expression patterns of enzymes e.g. RDH10 is quite varied in expression between tissues and is very high in the adult in regions such as retina and much lower in e.g. the brain and a number of blood cell types. Is retinaldehyde toxic? It was once an explanation why so little can be detected because it is a short-lived intermediate.  Is this the case? Line 97-98 described that “RDH10-deficient embryos displayed defects in axial extension and turning…” What is meant by turning? The subcellular distribution of a number of enzymes is described and figure 4 is particularly useful but the proposed expression of Rdh10 in mitochondria or lipid drops is not mentioned. Line 430 “their physiological functions are remarkably different and not overlapping.” Is it remarkable or can it just be said that they are different and not overlapping?

Typos

Line

41        this assay allowed tracking of the changes in RA

50        was caused by the severely reduced ability of mutant RDH10

66        The short-chain dehydrogenase/reductase (SDR) superfamily

103      another member of the SDR16C family

158      The activity of the RDH10 component of the ROC

303      as well as mice fed a vitamin A-deficient (VAD) diet.

471      the mouse enzyme was shown

473      a proposed unconventional mechanism of chromophore regeneration

Reviewer 3 Report

This review provides a substantial overview of the role and biochemistry of the various retinol dehyrogenases during the regulation of RA. Overall, it is beneficial for the RA community. Given that much of this work was conducted by the Kedishvili lab, who have authored this manuscript, it presents an exhaustive and state-of-the-art review on the topic. There are however a few issues that should be addressed in order to increase the clarity and accessibility of this publication:

Several grammatical issues, predominantly forgetting to use articles/determiners like “the” are found throughout the manuscript:

Line 37 add “the” in front of “gene encoding”

Line 41 add “for” in front of “tracking”

Line 54 add “the” in front of “Rdh10”

Line 55 “embryos” instead of “the embryos”

Line 57 remove “the” in front of “RA”

Line 66 add “The” to beginning of the sentence

Line 264. “Rdhe2” should be lower case

Line 268 italicize “rdhe2”

Lines 334-354 are the only ones presented in the present tense, this should be switched to past for consistency.

Line 339 replace “an” with “the”

Line 340 add “the” in front of “transcriptional”

Line 368 replace “transcript” with “transcription”

Line 375 replace “RDH11” with “RDH11’s”

Line 376 pluralize “chromophore”

Line 489 add “the” in front of “liver”

Line 462 remove “the” in front of “carbon”

Line 490 add “The” in front of “discovery”

Line 505, remove comma in front of  “RDH13”, add a verb in front of “to” (evaluate, uncover)

Line 513 add “the” in front of “National”

ROC should be defined in Figure 1, as in other figures. Furthermore, this is the first mention of the Retinoid oxidoreductive complex in the text. Line 81, was the website a placeholder for the authors to remember to include citation 16? It is unclear why it is included. While “paradoxical components” are philosophically interesting, a paragraph (lines 185-191) does not need to be utilized to explain this concept. The authors already provide an excellent synopsis of the unique features imparted by the formation of the ROC complex in line 192-199, which uncouples concentration dependence of either component. Simply citing the concept of paradoxical components (Ref 37) in line 192 is thus sufficient. Moreover, the authors appear to attribute a statement that “examples of paradoxical components existing in mammalian cells have been scarce” to Ref 34.  However, as noted by this review there are plenty of examples of known paradoxical components in mammals, eg. Notch/Delta, morphogens, bifunctional circuits, etc. While many of these were first reported in Drosophila/aves, these are well known interactions in mammals, for example Hart and Alon directly cite a report of PFK2 in mammalian metabolism from 1983. Lines 287-288. These findings reinforce the notion that this is a highly-specific It does not suggest anything about dependence/independence. The work in Ref 31 points to this latter point though. Neither Figure 4 nor Table 2 are cited in the text. This is easily resolvable for the table. Figure 4 does not have context for its current placement. Perhaps it would fit best early on as an introductory figure with a short section highlighting the general synthesis of RA and the topics to be discussed below.

Reviewer 4 Report

Dear authors,

In the manuscript by Belyaeva et al., you summarize the current state of knowledge regarding the identities of physiologically relevant retinol dehydrogenases, their enzymatic properties and tissue distribution, and discuss the potential mechanisms for the regulation of the flux from retinol to retinaldehyde.

I consider this work interesting and well organized with simple but explanatory figures.

The paragraphs are well organized.

The paragraph 7. underlines the importance that future studies are necessary for example to evaluate the contribution of RDH11-related enzymes, RDH13 and RDH14, to retinoid metabolism in adult and embryonic tissues.

I think that this manuscript should be accepted to be published in this journal.
